# Individual Keratinocyte Necroses in the Epidermis Are Apoptotic Keratinocytes in the Skin

**DOI:** 10.3390/diagnostics13223405

**Published:** 2023-11-08

**Authors:** Mitsuhiro Tachibana, Hideki Hamayasu, Kazuki Tomita, Yuta Kage

**Affiliations:** 1Department of Diagnostic Pathology, Shimada General Medical Center, Shimada 427-8502, Japan; hamayasu@shimada-gmc.jp; 2Department of Dermatology, Shimada General Medical Center, Shimada 427-8502, Japan; tommy1789vyu@gmail.com (K.T.); yutakage@gmail.com (Y.K.)

**Keywords:** keratinocyte necrosis, apoptosis, cleaved caspase-3, skin

## Abstract

The patient was a 44-year-old woman with Stevens–Johnson syndrome due to receiving Baktar^®^ (sulfamethoxazole trimethoprim) medication at our outpatient dermatology clinic. The epidermis, dermis, and subcutaneous adipose tissue samples showed numerous necrotic keratinocytes in the epidermis. Apoptotic nuclei were visualized as diaminobenzidine brown deposits with immunoperoxidase staining for cleaved caspase-3. The cleaved caspase-3-positive findings were consistent with eosinophilic material that appeared to be necrotic cells within the epidermis. Therefore, these eosinophilic materials may be apoptotic bodies. Generally speaking, eosinophilic cells are considered necrotic keratinocytes, especially in Japan. To the best of our knowledge, no studies have used apoptotic immunohistochemical markers to examine whether these structures are apoptotic in a human specimen.

Stevens–Johnson syndrome is a widespread erythema multiforme with ocular and mucosal lesions and systemic symptoms. It is most commonly caused by drugs. Histologically, epidermal necrotic keratinocytes or individual cell keratinization are observed, along with liquefaction of the basal layer and edema of the dermis. In the present case, we report epidermal keratinocyte changes in skin tissue from a patient with Stevens–Johnson syndrome using cleaved caspase-3 immunohistochemical staining.

The patient was a 44-year-old woman with Stevens–Johnson syndrome that developed due to Baktar^®^ (sulfamethoxazole trimethoprim) (Shionogi, Osaka, Japan) medication received at our outpatient dermatology clinic.

The patient presented with sores on her lips and mouth (Figure 1a) and erythema of the whole body (mainly on the palms and soles). A 5 mm skin punch biopsy was taken from erythema on the dorsal surface of her back (Figure 1b). The histopathological examination of the epidermis, dermis, and subcutaneous adipose tissue samples showed numerous necrotic keratinocytes in the epidermis (Figure 1c, arrows). In a recent study, the cleaved caspase-3 immunohistochemistry showed that a regulator of extracellular matrix (ECM) integrity, lumican, a small leucine-rich proteoglycan, exhibited anti-tumor properties in melanoma [1]. Therefore, cleaved caspase-3 immunohistochemistry is a very useful technique for studying dermatopathology.

Apoptotic nuclei were visualized as diaminobenzidine brown deposits with immunoperoxidase staining for cleaved caspase-3 by using a 1:500-diluted rabbit polyclonal antibody available from Cell Signaling Technology (Danvers, MA, USA) as previously described [2]. The cleaved caspase-3-positive findings were consistent with eosinophilic materials that appeared to be necrotic cells within the epidermis (Figure 1d). Therefore, these eosinophilic materials may be apoptotic bodies. The patient was treated with oral prednisolone and the erythema faded and healed. Dermatological treatment was completed nine months after the onset of the rash.

Generally speaking, and especially in Japan, these materials are considered necrotic keratinocytes. That is to say, the eosinophilic cells are considered dyskeratotic cells or necrotic keratinocytes, not apoptotic keratinocytes in Japan [3]. However, McKee’s Pathology of the Skin and Weedon’s Skin Pathology [4,5] show these materials as apoptotic keratinocytes. Programmed cell death, or apoptosis, is an intracellular mechanism essential for homeostasis in multicellular organisms and is widely used to eliminate unwanted cells, such as damaged cells [6]. However, apoptosis is no longer synonymous with programmed cell death, as other forms of programmed death, such as necroptosis and pyroptosis, have been identified. Each death depends on a different mechanism and has different consequences in vivo. Apoptosis is generally accepted as a programmed cell death mechanism that does not cause inflammation [6]. Necroptosis and pyroptosis, however, are inflammatory deaths characterized by cell swelling, formation of membrane pores, and rupture of the cell membrane [6]. Thus, both necroptosis and pyroptosis result in the release of inflammatory intracellular contents and cause inflammation. However, they have distinct functions and signaling pathways [6]. Necroptosis is often observed as a backup system initiated when apoptosis is inhibited. In contrast, pyroptosis is a primary cellular response mediated by the inflammasome after sensing extensive PAMPs and DAMPs [6].

Apoptotic cell death can be divided into two major pathways: (1) the intrinsic pathway, which is activated by cellular stress or injury, and (2) the extrinsic pathway, which is initiated by the triggering of death receptors. Both pathways lead to activation of effector caspases such as caspase-3, -6, and -7, resulting in apoptosis [6]. Keratinocyte death in Stevens–Johnson syndrome occurs by apoptosis. There are two pathways leading to apoptotic cell death: the ligation of Fas by its ligand FasL and the release of perforin/granzyme B. Keratinocyte apoptosis is mediated by the ligation of Fas on keratinocytes by FasL on T cells [7]. Therefore, the term keratinocyte “necrosis” in Stevens–Johnson syndrome is incorrect terminology. In this paper, the most important thing is that the term keratinocyte necrosis, used by Japanese researchers, has little basis in medical biology and is essentially apoptosis.

To the best of our knowledge, no studies have used apoptotic immunohistochemical markers to examine whether these structures are apoptotic in a human specimen. Further multicenter studies with more cases are warranted.

## Figures and Tables

**Figure 1 diagnostics-13-03405-f001:**
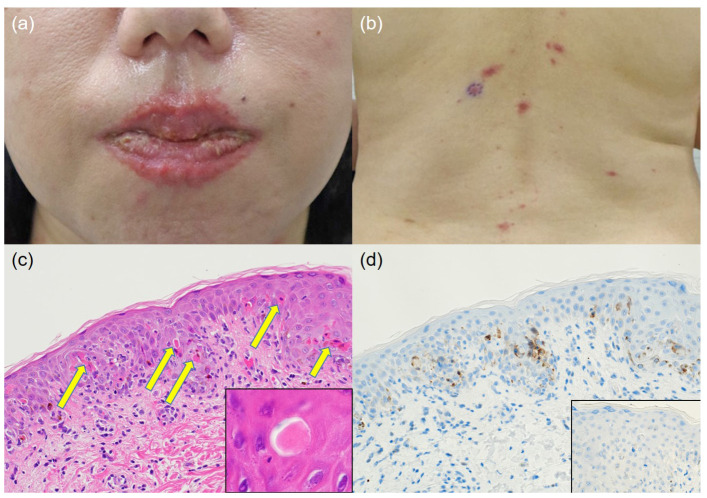
(**a**) Macroscopic findings of the rash on the lip. (**b**) Macroscopic findings of the rash on the back. (**c**) Microscopic findings. The epidermis shows numerous eosinophilic unstructured necrotic keratinocytes (×200) (arrows, inset (×1000)) with exocytosis of small lymphocytes (hematoxylin and eosin staining). (**d**) The cleaved caspase-3-positive findings are consistent with eosinophilic amorphous material that appeared to be necrotic cells within the epidermis (×200) (cleaved caspase-3 immunostaining; inset: negative control staining).

## Data Availability

Data is contained within the article.

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
