# Peer review of "Individual Keratinocyte Necroses in the Epidermis Are Apoptotic Keratinocytes in the Skin"

_diagnostics, 2023, doi:10.3390/diagnostics13223405_

Round 1
Reviewer 1 Report
Comments and Suggestions for Authors
For several years it has been known that in pathologies associated with the skin, keratinocytes die by various means including necrosis and apoptosis.
Gandarillas A, Goldsmith LA, Gschmeissner S, Leigh IM, Watt FM. Evidence that apoptosis and terminal differentiation of epidermal keratinocytes are distinct processes. Exp Dermatol. 1999 Feb;8(1):71-9. doi: 10.1111/j.1600-0625.1999.tb00350.x. PMID: 10206724.
Wrone-Smith T, Mitra RS, Thompson CB, Jasty R, Castle VP, Nickoloff BJ. Keratinocytes derived from psoriatic plaques are resistant to apoptosis compared with normal skin. Am J Pathol. 1997 Nov;151(5):1321-9. PMID: 9358758; PMCID: PMC1858068.
Therefore, the present clinical case does not show relevant information.
The authors mention that there are no studies that use apoptosis markers to examine these structures in keratinocytes, which is not correct since there are studies in this regard.
Tobón-Arroyave SI, Villegas-Acosta FA, Ruiz-Restrepo SM, Vieco-Durán B, Restrepo-Misas M, Londoño-López ML. Expression of caspase-3 and structural changes associated with apoptotic cell death of keratinocytes in oral lichen planus. Oral Dis. 2004 May;10(3):173-8. doi: 10.1046/j.1601-0825.2003.00998.x. PMID: 15089928.
Elsherif R, Mahmoud WA, Mohamed RR. Melanocytes and keratinocytes morphological changes in vitiligo patients. A histological, immunohistochemical and ultrastructural analysis. Ultrastruct Pathol. 2022 Mar 4;46(2):217-235. doi: 10.1080/01913123.2022.2044946. Epub 2022 Mar 4. PMID: 35243959.
In general, a relevant contribution is not demonstrated with the presentation of the clinical case, the writing is more focused on the use of the immunohistochemistry technique than on the description and findings of the clinical case.
Author Response
For several years, it has been known that in pathologies associated with the skin, keratinocytes die by various means, including necrosis and apoptosis.
Gandarillas A, Goldsmith LA, Gschmeissner S, Leigh IM, Watt FM. Evidence that apoptosis and terminal differentiation of epidermal keratinocytes are distinct processes. Exp Dermatol. 1999 Feb;8(1):71-9. doi: 10.1111/j.1600-0625.1999.tb00350.x. PMID: 10206724.
Wrone-Smith T, Mitra RS, Thompson CB, Jasty R, Castle VP, Nickoloff BJ. Keratinocytes derived from psoriatic plaques are resistant to apoptosis compared with normal skin. Am J Pathol. 1997 Nov;151(5):1321-9. PMID: 9358758; PMCID: PMC1858068.
Therefore, the present clinical case does not show relevant information.
Thank you for your comment. When describing keratinocyte cell death in dermatopathology, Japanese dermatologists and pathologists usually use "necrosis." In contrast, English books often use the word "apoptosis," and I wanted to clarify this and show the problem.
The authors mention that there are no studies that use apoptosis markers to examine these structures in keratinocytes, which is not correct since there are studies in this regard.
Tobón-Arroyave SI, Villegas-Acosta FA, Ruiz-Restrepo SM, Vieco-Durán B, Restrepo-Misas M, Londoño-López ML. Expression of caspase-3 and structural changes associated with apoptotic cell death of keratinocytes in oral lichen planus. Oral Dis. 2004 May;10(3):173-8. doi: 10.1046/j.1601-0825.2003.00998.x. PMID: 15089928.
Elsherif R, Mahmoud WA, Mohamed RR. Melanocytes and keratinocytes morphological changes in vitiligo patients. A histological, immunohistochemical, and ultrastructural analysis. Ultrastruct Pathol. 2022 Mar 4;46(2):217-235. doi: 10.1080/01913123.2022.2044946. Epub 2022 Mar 4. PMID: 35243959.
Thank you for your comment. I searched only the literature on skin and did not look at the literature related to oral pathology. I will add it to the references.
In general, a relevant contribution is not demonstrated with the presentation of the clinical case, the writing is more focused on the use of the immunohistochemistry technique than on the description and findings of the clinical case.
Thank you for pointing this out. I have described the immunohistochemical search in detail to present whether the death of keratinocytes is apoptosis or necrosis.
Reviewer 2 Report
Comments and Suggestions for Authors
Please expand a little the introduction and the followup of the case
Please correct the sentence "Therefore, these eosinophilic materials 43 must be apoptotic bodies.", I feel that "must" is not appropriate.
Better explain what you mean with "Generally speaking, and especially in Japan, these materials 44 are considered necrotic keratinocytes"
I think it would be of interest to report on other use of IHC for cleaved-caspase3 in skin pathology, with more in depth in the literature
Comments on the Quality of English LanguageSee above
Author Response
Please expand a little the introduction and the follow-up of the case
Thank you for your comment. We have added the following information.
Introduction
Stevens-Johnson syndrome is defined as widespread erythema multiforme with ocular and mucosal lesions and systemic symptoms. It is most commonly caused by drugs. Histologically, epidermal necrotic keratinocytes or individual cell keratinization are observed, along with liquefaction of the basal layer and edema of the dermis. In the present case, we report on epidermal keratinocyte changes in skin tissue from a patient with Stevens-Johnson syndrome, using cleaved caspase-3 immunohistochemical staining.
Follow-up information
The patient was treated with oral prednisolone, and the erythema faded and healed. Dermatological treatment was completed 9 months after the onset of the rash.
Please correct the sentence "Therefore, these eosinophilic materials 43 must be apoptotic bodies.", I feel that "must" is not appropriate.
We have corrected it, "Therefore, these eosinophilic materials may be apoptotic bodies."
Better explain what you mean with "Generally speaking, and especially in Japan, these materials are considered necrotic keratinocytes"
Thank you for your comment. We have added the following information.
That is to say, the eosinophilic cells are considered dyskeratotic cells or necrotic keratinocytes, not apoptotic keratinocytes in Japan,
I think it would be of interest to report on other use of IHC for cleaved-caspase3 in skin pathology, with more in depth in the literature
Thank you for your comment. We have added the following information.
In recent study, the cleaved caspase-3 immunohistochemistry showed that a regulator of extracellular matrix (ECM) integrity, lumican, a small leucine-rich proteoglycan, exhibited anti-tumor properties in melanoma. Therefore, the cleaved caspase-3 immunohistochemistry is a very useful technique for studying dermatopathology.
Ref) Brézillon S, Untereiner V, Mohamed HT, Ahallal E, Proult I, Nizet P, Boulagnon-Rombi C, Sockalingum GD. Label-free infrared spectral histology of skin tissue Part II: Impact of a lumican-derived peptide on melanoma growth. Front Cell Dev Biol. 2020; 8:377.
Reviewer 3 Report
Comments and Suggestions for Authors
The authors describe a case report, where they use histological staining on a Stevens-Johnson syndrome patient.
They complete HE staining claiming for necrotic kertainocytes and casp-3 staining to show apoptotic cells, which the claim overlap.
For the ease of understanding, I would suggest to use arrows to highlight the cells the authors are referring to as eosinophylic cells. To also claim that casp-3 positive cells are eosinophylic, I would suggest to do a double staining ont he samples, if possible.
The authors also write: „As reported previously, necrotic keratinocytes can also be a possibility.” but is it is previously reported, then the citation should be included.
Although, for years the different cell death methods were considered very distinct, nowadays it seems that apoptosis, necrosis and pyroptosis are not mutually exclusive and can be present at the same time with some overlaps, i.e. necroptosis etc. I think this could be discussed a bit in the manuscript.
Author Response
The authors describe a case report, where they use histological staining on a Stevens-Johnson syndrome patient.
They complete HE staining claiming for necrotic keratinocytes and casp-3 staining to show apoptotic cells, which the claim overlap.
For the ease of understanding, I would suggest to use arrows to highlight the cells the authors are referring to as eosinophylic cells. To also claim that casp-3 positive cells are eosinophylic, I would suggest to do a double staining on the samples, if possible.
The authors also write: „As reported previously, necrotic keratinocytes can also be a possibility.” but is it is previously reported, then the citation should be included.
Thank you for the detailed advice, I pointed out the eosinophilic cells in the HE-stained image with an arrow. Double staining is not available at our institution. Instead, we performed a negative control stain to show that the cleaved caspase-3 stain is not a nonspecific finding. We also could not find any previous reports of possible necrotic keratinocytes in the literature and removed the statement "As reported previously, necrotic keratinocytes can also be a possibility ." was removed.
Although, for years the different cell death methods were considered very distinct, nowadays it seems that apoptosis, necrosis and pyroptosis are not mutually exclusive and can be present at the same time with some overlaps, i.e. necroptosis etc. I think this could be discussed a bit in the manuscript.
Thank you for your comment. We have added the following information.
Programmed cell death, or apoptosis, is an essential intracellular mechanism for maintaining homeostasis in multicellular organisms, and is widely used for removing unwanted cells, such as damaged cells. However, apoptosis is no longer solely synonymous with programmed cell death, because of the identification of other forms of programmed death, such as necroptosis and pyroptosis. Each death depends on different mechanisms and yields different results in vivo. Apoptosis is generally accepted as a programmed cell death machinery, which essentially does not elicit inflammation. However, necroptosis and pyroptosis are inflammatory types of death that are characterized by cell swelling, membrane pore formation, and plasma membrane rupture. Therefore, both necroptosis and pyroptosis result in the release of inflammatory intracellular contents, leading to inflammation. However, they have distinct functions and signaling pathways. While necroptosis is mostly observed as a back-up system that is initiated when apoptosis is blocked, pyroptosis is an inflammasome-mediated primary cellular response following the sensing of a broad range of PAMPs and DAMPs. Apoptotic cell death has been divided into two broad categories: (1) the intrinsic pathway, which is activated by cellular stress and injury, and (2) the extrinsic pathway, which is initiated by the triggering of death receptors. Both pathways lead to the activation of effector caspases, such as caspase-3, -6, and -7, resulting in apoptosis.
Ref) Lee K-H, Kang T-B. The molecular links between cell death and inflammasome. Cells. 2019; 8:1057.
Reviewer 4 Report
Comments and Suggestions for Authors
Apoptosis is a type of necrosis mediated by peculiar intracelular pathways. Despite it is nice to see that the apoptotic keratinocyes in SSJ stained with the marker, this does not change either the diagnosis or other significant approach.
Comments on the Quality of English LanguageOk
Author Response
Thank you for your comment. Apoptosis is not a type of necrosis. In this study, we found that Japanese dermatologists/pathologists and Western dermatopathologists disagree on whether the death of epidermal keratinocytes in SJS is apoptosis or necrosis, and we performed cleaved caspase-3 immunohistochemistry staining to show which argument is correct. This is the first time that a skin sample has been used in this way. This is an unprecedented study on skin specimens and is worth presenting.
Reviewer 5 Report
Comments and Suggestions for Authors
The title is unclear and should be changed to make it clear. I believe you want to say ‘Individual Keratinocyte Necrosis in the Epidermis are Apoptotic Keratinocytes’? please make this clear. Explaining clearly what was the purpose of this observation would allow a better understanding of the article.
Explain why you applied Cleaved-caspase 3 immunostaining - is this a recognized marker of apoptotic nuclei?
Necrotic material may uptake immunoreactants in a non-specific manner, and therefore the immunostaining shown in the figure could be non-specific. Please provide an image of a negative control (the same procedure but omitting application of the antibody).
Comments on the Quality of English LanguageThe English needs thorough revision. Eg. ‘the dorsal surface of her back’ is a pleonasm, the back has only one surface. ‘Dermal punch biopsy’ should read ‘Skin punch biopsy’. ‘…erythema of the whole body’ is not accurate (erythema of the whole body means erythroderma, which this patient has not). ‘To our best knowledge’ should read ‘To the best of our knowledge’. ‘Suspected’ Stevens-Johnson syndrome’ is not a satisfactory diagnosis. If the diagnosis of Stevens-Johnson syndrome is certain, the term ‘suspected’ should be omitted. Otherwise please provide a convincing diagnosis.
Author Response
Comments and Suggestions for Authors
The title is unclear and should be changed to make it clear. I believe you want to say ‘Individual Keratinocyte Necrosis in the Epidermis are Apoptotic Keratinocytes’? please make this clear. Explaining clearly what was the purpose of this observation would allow a better understanding of the article.
Thank you for your comment. The title of this manuscript was changed to “Individual Keratinocyte Necrosis in the Epidermis are Apoptotic Keratinocytes in the Skin: A Case Report.”
Explain why you applied Cleaved-caspase 3 immunostaining - is this a recognized marker of apoptotic nuclei?
Thank you for pointing out that Cleaved-caspase 3 immunostaining was used as a histological marker for apoptotic nuclei.
I have described Cleaved-caspase 3 below.
Cleaved caspase-3, representing an activated form of caspase-3, is an excellent and reproducible immunohistochemical marker for apoptosis [1,2]. The biochemical pathways of apoptosis are controlled by caspases (cysteine aspartate-specific proteases), which cleave and activate a variety of intracellular proteins [3,4]. Cleaved caspase 3 functions as a kind of control tower of apoptosis: it cleaves poly (ADP-ribose) polymerase (PARP), cytokeratin 18, vimentin, actin and other intracytoplasmic proteins. Cleaved caspase 3 has been applied to detecting apoptotic neoplastic cells in paraffin sections [5-7].
- Panicker NK, Buch AC, Patel AR: Breast carcinoma with numerous large "thanatosomes". J Cancer Res Ther. 2015, 11:980-2.
- Gown AM, Willingham MC: Improved detection of apoptotic cells in archival paraffin sections: immunohistochemistry using antibodies to cleaved caspase 3. J Histochem Cytochem. 2002, 50:449-54.
- Tsutsumi Y, Kamoshida S: Pitfalls and caveats in histochemically demonstrating apoptosis. Acta Histochem Cytochem. 2003, 36:271-284.
- Thornberry NA, Rano TA, Peterson EP, et al.: A combinatorial approach defines specificities of members of the caspase family and granzyme B. Functional relationships established for key mediators of apoptosis. J Biol Chem. 1997, 272:17907-11.
- Leite AF, Bernardo VG, Buexm LA, Fonseca EC, Silva LE, Barroso DR, Lourenço Sde Q: Immunoexpression of cleaved caspase-3 shows lower apoptotic area indices in lip carcinomas than in intraoral cancer. J Appl Oral Sci. 2016, 24:359-65.
- Liu PF, Hu YC, Kang BH, et al.: Expression levels of cleaved caspase-3 and caspase-3 in tumorigenesis and prognosis of oral tongue squamous cell carcinoma. PLoS One. 2017, 12:e0180620.
- Fang Y, Li J, Wu Y, Gui J, Shen Y: Costunolide Inhibits the Growth of OAW42-a multidrug-resistant human ovarian cancer cells by activating apoptotic and autophagic pathways, production of reactive oxygen species (ROS), cleaved caspase-3 and cleaved caspase-9. Med Sci Monit. 2019, 25:3231-7.
Necrotic material may uptake immunoreactants in a non-specific manner, and therefore the immunostaining shown in the figure could be non-specific. Please provide an image of a negative control (the same procedure but omitting application of the antibody).
Thank you for your comment. I have presented a negative control image.
Comments on the Quality of English Language
The English needs thorough revision. Eg. ‘the dorsal surface of her back’ is a pleonasm, the back has only one surface. ‘Dermal punch biopsy’ should read ‘Skin punch biopsy’. ‘…erythema of the whole body’ is not accurate (erythema of the whole body means erythroderma, which this patient has not). ‘To our best knowledge’ should read ‘To the best of our knowledge’. ‘Suspected’ Stevens-Johnson syndrome’ is not a satisfactory diagnosis. If the diagnosis of Stevens-Johnson syndrome is certain, the term ‘suspected’ should be omitted. Otherwise please provide a convincing diagnosis.
We have made the corrections as you indicated. In addition, we have proofread the English text.
Round 2
Reviewer 1 Report
Comments and Suggestions for Authors
The authors made important changes to the manuscript, which is reflected in the presentation of the results.
The manuscript has better order and clarity in its presentation.
May be accepted for publication.
Author Response
The authors made important changes to the manuscript, which is reflected in the presentation of the results.
The manuscript has better order and clarity in its presentation.
May be accepted for publication.
Thank you for your comments. Your comments make me happy.
Reviewer 2 Report
Comments and Suggestions for Authors
I feel that discussion and comparison with literature is still a bit short, and I would suggest more expansion; otherwise I have no other comments.
Comments on the Quality of English LanguageAs a not-mothertongue, I can feel the difficulties and I have no major comments.
Author Response
I feel that discussion and comparison with literature is still a bit short, and I would suggest more expansion; otherwise I have no other comments.
Thank you for your comments. The following document has been added.
Keratinocyte death in Stevens-Johnson syndrome occurs by apoptosis. There are two pathways leading to apoptotic cell death, the ligation of Fas by its ligand FasL and the release of perforin/granzyme B. Keratinocyte apoptosis is mediated by the ligation of Fas on keratinocytes by FasL on T cells [7]. Therefore, we think that the term keratinocyte “necrosis” in Stevens-Johnson syndrome is incorrect terminology.
7. Borchers, A.T.; Lee, J.L.; Naguwa, S.M.; Cheema, G.S.; Gershwin, M. E. Stevens-Johnson syndrome and toxic epidermal necrolysis. Autoimmun Rev 2008,7,598-605.
Reviewer 5 Report
Comments and Suggestions for Authors
Thank you for having replied to the reviewers' comments. I believe your text is sill not very clear as to what the purpose of the study was.
Comments on the Quality of English LanguageEnglish is stil (very) poor and needs significant improvement.
Author Response
Thank you for having replied to the reviewers' comments. I believe your text is sill not very clear as to what the purpose of the study was.
Thank you for your comments. The following document has been added. “In this paper, the most important point the authors wish to make is that the term keratinocyte necrosis, as used by Japanese researchers, has little basis in medical biology and is essentially apoptosis.”